# Preventing Evaporation Products for High-Quality Metal Film in Directed Energy Deposition: A Review

**Kang-Hyung Kim** [1] [ID]**, Chan-Hyun Jung** [2]**, Dae-Yong Jeong** [2] **and Soong-Keun Hyun** [1,2,*]

1   Program in Metals and Materials Process Engineering, Inha University, Incheon 22212, Korea;
    ds2pav@hotmail.com
2   Department of Materials Science and Engineering, Inha University, Incheon 22212, Korea;
    changus0100@gmail.com (C.-H.J.); dyjeong@inha.ac.kr (D.-Y.J.)
*   Correspondence: skhyun@inha.ac.kr; Tel.: +82-32-860-7547; Fax: +82-32-862-5546

**Abstract:** Directed energy deposition (DED), a type of additive manufacturing (AM) is a process that enables high-speed deposition using laser technology. The application of DED extends not only to 3D printing, but also to the 2D surface modification by direct laser-deposition dissimilar materials with a sub-millimeter thickness. One of the reasons why DED has not been widely applied in the industry is the low velocity with a few m/min, but thin-DED has been developed to the extent that it can be over 100 m/min in roller deposition. The remaining task is to improve quality by reducing defects. Thus far, defect studies on AM, including DED, have focused mostly on preventing pores and crack defects that reduce fatigue strength. However, evaporation products, fumes, and spatters, were often neglected despite being one of the main causes of porosity and defects. In high-quality metal deposition, the problems caused by evaporation products are difficult to solve, but they have not yet caught the attention of metallurgists and physicists. This review examines the effect of the laser, material, and process parameters on the evaporation products to help obtain a high-quality metal film layer in thin-DED.

**Keywords:** cavitation bubble; keyhole; nanoparticles; fume; spatter

## 1. Introduction

Directed energy deposition (DED) is an additive manufacturing technology (AM) used to manufacture 3D structures. The process can also be used to produce 2D deposition with dissimilar material deposition technology that forms a sub-millimeter coated layer. Thin-DED is particularly suitable to replace hexavalent chromium plating [1,2], which is very harmful to the human body. Among the other alternative techniques, plasma spray and high-velocity oxygen fuel (HVOF) spray have been strong candidates to replace chromium plating for over twenty years, but they are still incomplete because of low bonding strength. Besides, flame spraying with fusing or detonation-gun (D-gun) shows the strength of metallurgical bonding, but they are difficult to replace in the industry due to the expensive process cost and difficulty of uniform and thin deposition. On the other hand, DED can achieve high bonding strength similar welding to fix delamination failure, which is persistent in plated and thermal sprayed layers, and also has an important industrial significance owing to the high degree of freedom in material selection and deposited thickness.

Common defects in powder AM include porosity due to LOF (lack of fusion) or gases and cracks caused by differences in thermal expansion during cooling [3–7]. On the other hand, it is difficult to manufacture porosity-free or crack-free products in the manufacturing stage using current technologies. Efforts to detect defects during the process or after completion by non-destructive testing, such as XCT (X-ray computed tomography) [3,5] and acoustic emission [4], are continuing. In the DED process, a metal powder or wire is supplied while focusing the laser on the base metal surface to form a

melt pool by a reaction with the laser light and laminate it to a height of several millimeters. In that respect, it is similar to cladding [8,9].

To replace plating or thermal spray coating, it is necessary to deposit in a thin layer while preserving the original shape of the base metal, which is difficult using the general DED method. Figure 1 presents a schematic diagram comparing PBF (Powder Bed Fusion) and DED, which are typical processes in metal AM, with thin-DED. Figure 1a is PBF, which is formed by laser irradiation of a thin layer of metal powder and melting the desired area layer by layer, and Figure 1b is DED, which produces a melt pool with a laser beam on the base metal, followed by the supply of a metal wire or powder to the melt pool with deposition in millimeter units. Figure 1c is a thin-DED that combines the advantages of both processes. In thin-DED, while the incident laser beam supplies powder to a two-dimensional surface of the workpiece, a thin layer of 50–300 micrometers is deposited per scan. Unlike general DED, thin-DED has a unique characteristic that the base metal is barely diluted. Only the powder is fused, so there is virtually no diluted bonding layer or little penetration.

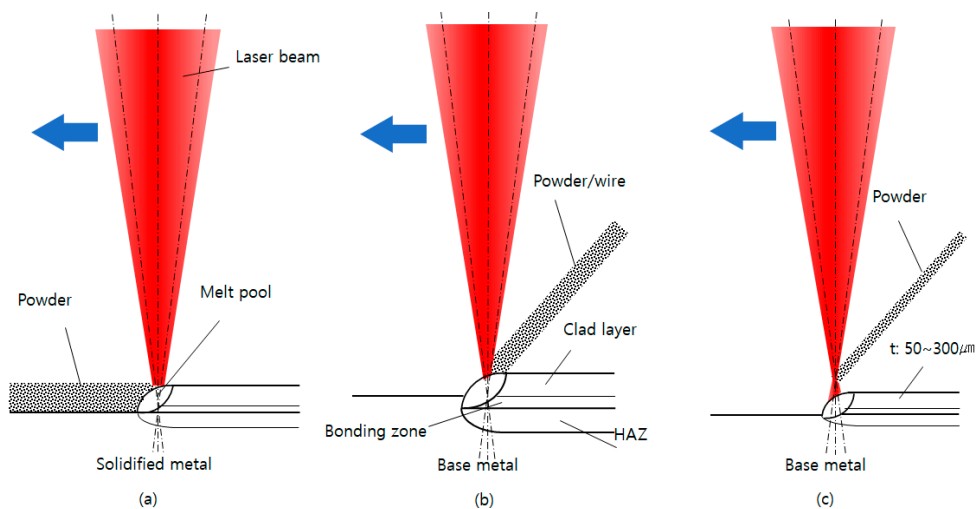

**Figure 1.** Schematic diagrams of metal AM processes: (**a**) PBF; (**b**) DED; (**c**) thin-DED.

In the general DED process, when the evaporated metal expands to form bubbles, the internal vapor pressure of the bubbles increases, and the bubbles burst to release evaporation products. The volatilized nanoparticles of evaporated metal are generally called a fume [10]. The massive particles resulting from the agglomeration of evaporated nanoparticles to a size of several hundred micrometers are called spatter [11–13]. Fumes are small enough to float in the atmosphere or hinder the reaction between the material and the laser and have very high reactivity with the human body or melt. Therefore, it harms the health of workers [14] and degrades the quality of the deposited layer due to a photochemical reaction with the laser, porosity, change in chemical composition, and impurities. Spatter contaminates tracks on the workpiece, causing various defects, such as increased roughness, porosity, contamination of the nozzle surface, decreased fatigue strength, decreased thermal conductivity, and low fluidity. In the general DED or cladding process, the solidification time is long, so there is some time for the dissolved gas absorbed in the melt to release. Since the melt pool is deep and wide, any spatter attached to the surface is also remelted, so there are few problems caused by evaporation products. In thin-DED, however, the time to solidify the deposited layer is very short so that fumes can become trapped in the melt and produce pores. Furthermore, pores can be formed in the gaps as molten powders are again laminated on the massive spatter.

In this study, to manufacture a high-quality metal thin film by a thin-DED process, the mechanism of forming evaporation products by a laser, the reaction with the laser according to the properties of powder materials, and the influence of the process parameters

were investigated in depth, shown in Table 1. By adjusting the laser density and process parameters based on these findings, it will be possible to prevent evaporation products and form a high-quality metal film layer in the thin-DED process.

**Table 1.** Preventive methods of fume and spatter in DED.

| Relevant DED Defects | Major Factors | Sub-Factors | Preventive Measures |
|---|---|---|---|
| Lower corrosion resistance, decreased fatigue strength, inner crack, surface crack, surface dimple, porosity, high roughness, uneven hardness, decreased laser energy efficiency, lower fluidity of melt. | Material | Absorptivity, small powder size (<15 μm), presence of low melting point elements or low vaporization temperature elements (<2000 °C), presence of carbon or boron, eutectic reaction, low wettability, characteristics of base metal. | Adjustment of laser power, irradiation angle, surface condition of the powder and base material. Appropriate choice of suitable powder size, addition of compound-forming refractory components (Nb, Ta, W, Zr). Replacement to very low carbon- and boron-contained powder. |
| | Source Laser | Laser type, laser power density, beam divergence, duration time, wavelength of laser. | Adjustment of laser power, powder feed rate. Selection of CW laser, top-hat mode beam, larger diameter beam, rectangular beam, or defocusing. |
| | Working Condition | Hatches distance, powder feed rate, scanning velocity, nozzle distance, cooling rate, layer thickness, humidity, vibration. | Redefining new conditions from single track experiments. Adjustment of hatches distance, powder feed rate, scanning velocity, nozzle distance, cooling rate, and layer thickness. Usage of air conditioner, and vibration absorber. |
| | Assist gas | Gas pressure, kind of gas. | Appropriate pressure for complete air shielding. Replacement of nitrogen or mixed argon-nitrogen gas to pure inert gas (argon or mixed argon-helium). |
| | Laser Focusing | Beam mode, beam shape, nozzle design, focal length, focused on the base material surface. | Appropriate choice of top-hat mode beam, larger diameter beam, rectangular beam, defocusing, inclined laser beam. |

## 2. Generation of Evaporation Products

### 2.1. Evaporation by Laser

In the past, fumes were understood as the vaporization of solid metals by high energy. In recent years, the basic mechanism of fume generation involves the evaporation of elements and oxides by plasma from a superheated melt. The fume is comprised of nanoparticles from the metallic vapor phase [15–17]. Metallic nanoparticle fumes are generated by various physicochemical methods, such as laser, electron beam, ion beam, plasma ionization, electromagnetic wave, arc discharge, combustion, spray pyrolysis, physical or chemical vapor deposition, but an analysis of the process is not simple.

Since Anisimov analyzed metal vapor motion in a vacuum by hydrodynamics in 1968 [18] molecular dynamics (MD), hydrodynamics (HD), and direct simulation Monte Carlo (DSMC) methods have been performed to understand the plasma formation and the plume behavior, making significant progress in comprehending the stages of evaporation [19–47]. Recently, the stage of the plume generation has been observed directly by shadowgraphy in laser confocal scanning microscopy (LCSM) [48,49] and high-speed scanning small-angle X-ray scattering (SAXS) [50,51].

The released fumes interrupt the laser energy transfer to the additive powder and base metal and often enter the deposited layer as impurities. Furthermore, agglomerated massive particles are released as spatter, contaminating the nozzle and the base metal surface.

*2.2. Evaporation Mechanism on Thin-DED Process*

In the thin-DED process, when high-speed photons hit the base metal surface as increased light intensity, free electrons of metal atoms absorb the energy to generate the laser-induced plasma [52]. At the focus of the laser beam, the base metal is heated rapidly to form a melt pool and releases nanoparticles to the atmosphere during bubble collapse through the bubble expansion and plasma plume stages [50]. When a keyhole forms in the melt pool, robust hydrodynamic melt flow rises vertically due to the recoil pressure and the Marangoni effect [45,53,54]. This results in the release of nanoparticles and several hundred micrometers sized spatter particles. All reactions from plasma formation to nanoparticle release occurred within only a few tens to hundreds of microseconds [55]. This occurs more frequently in pulsed-wave laser than continuous-wave laser because of the high peak power [56].

## 3. Influence of Laser Characteristics

The high energy laser is difficult to control in the DED process because it reacts with the deposition material and base metal within tens to hundreds of microseconds. Moreover, the loss of the delivery system, cooling system, optical lens, and conversion system vary greatly depending on the source laser. Furthermore, the efficiency varies greatly depending on the deposition material and the surface condition of the base metal. The shape of the deposition material is an important parameter. The drop size of the molten powder is finer than that of a molten wire drop, allowing uniform and thin deposition. General DED is melted by the photochemical reaction of a laser, base metal, and deposition material. Thin-DED involves a reaction mainly between the laser and the powder, minimizing melt pool formation on the base metal, resulting in a thin and uniform deposited layer. The thickness of the deposited layer obtained by thin-DED is 50–300 micrometers, which is quite thin, and there are little bonding zones caused by dilution. For this, cavitation bubbles and keyhole phenomena must be suppressed. Precise control of the powder feed rate from the nozzle, laser power adjustment, scanning velocity, and the location where the powder and laser meet are key parameters in the process.

*3.1. Laser-Induced Cavitation Bubble*

Cavitation is a phenomenon in which vapor bubbles are generated in a liquid when the saturated vapor pressure decreases with increasing temperature or velocity. In thin-DED, cavitation bubbles can be formed by the laser plasma when the fluid pressure or the fluid velocity of the melt increases rapidly [50].

Figure 2 presents a conceptual diagram showing how cavitation bubbles are generated and grown by laser irradiation, nanoparticles are released as fumes, and aggregated massive particles are released as spatter. Initially, a cavitation bubble, which is in the form of a nano-scale hemisphere (I), expands to a millimeter size [50]. The jet flow velocity of nanoparticles along the plasma plume rises perpendicularly to the surface of the base metal and passes through a conical shape (II) to a pointed top. It becomes form (III). Stauss et al. explained (Figure 3) the step-by-step process of emitting nanoparticles after the expansion of cavitation bubbles during laser ablation. Since the cavitation bubble interface is not a material film, it is formed only by the difference in pressure and density. Some nanoparticles may escape from the cavitation bubble into the surrounding fluid [46].

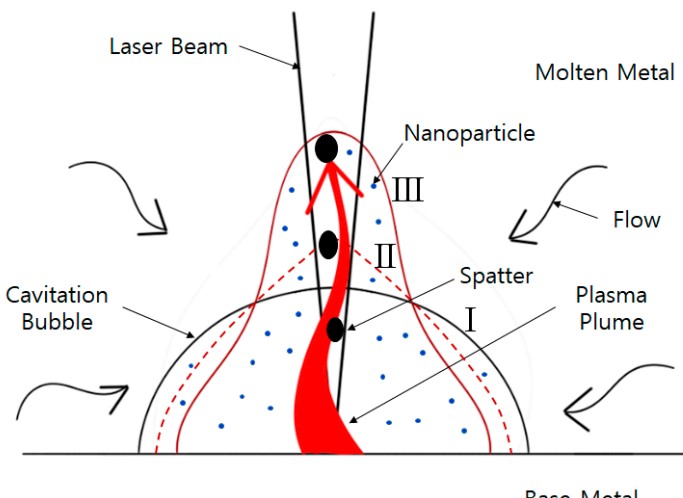

**Figure 2.** Modified Ibrahimkutty's conceptual diagram for fluid flow in the cavitation bubble, immediately before the generation of evaporation products, bubble stages: hemispherical cavitation bubble (I) expansion of a cone shape bubble (II, dotted line) ejection of a spin top shape bubble (III) [50].

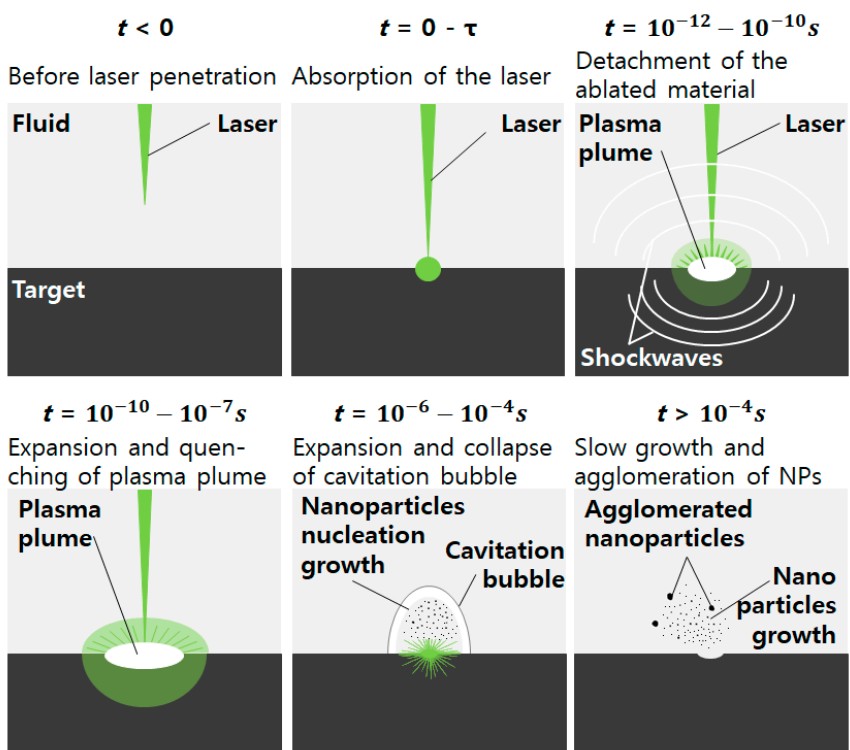

**Figure 3.** Rapid heating by a laser beam and subsequent plasma formation leads to the formation of cavitation bubbles, then vaporization by nanoparticle (NP) nucleation and growth in a pulsed laser, at t ~ $10^{-6}$–$10^{-4}$ s. (Modified from Stauss' nanomaterial nucleation diagram [46]).

Muneoka et al. described the following six stages in more detail: plasma formation (phase I), cavitation bubble expansion (phase II), bilayer cavitation bubble expansion (phase III), contraction (phase IV), stagnation (phase V), and cavitation bubble collapse with nanoparticle release (phase VI) [51]. Ibrahimkutty et al. observed the moments at 320 microseconds (phase VI) in which the bubble collapse after the rise of the plasma plume inside the cavitation bubble, as shown in Figure 4 by X-ray [50]. At that moment, the evaporation products are released into the atmosphere.

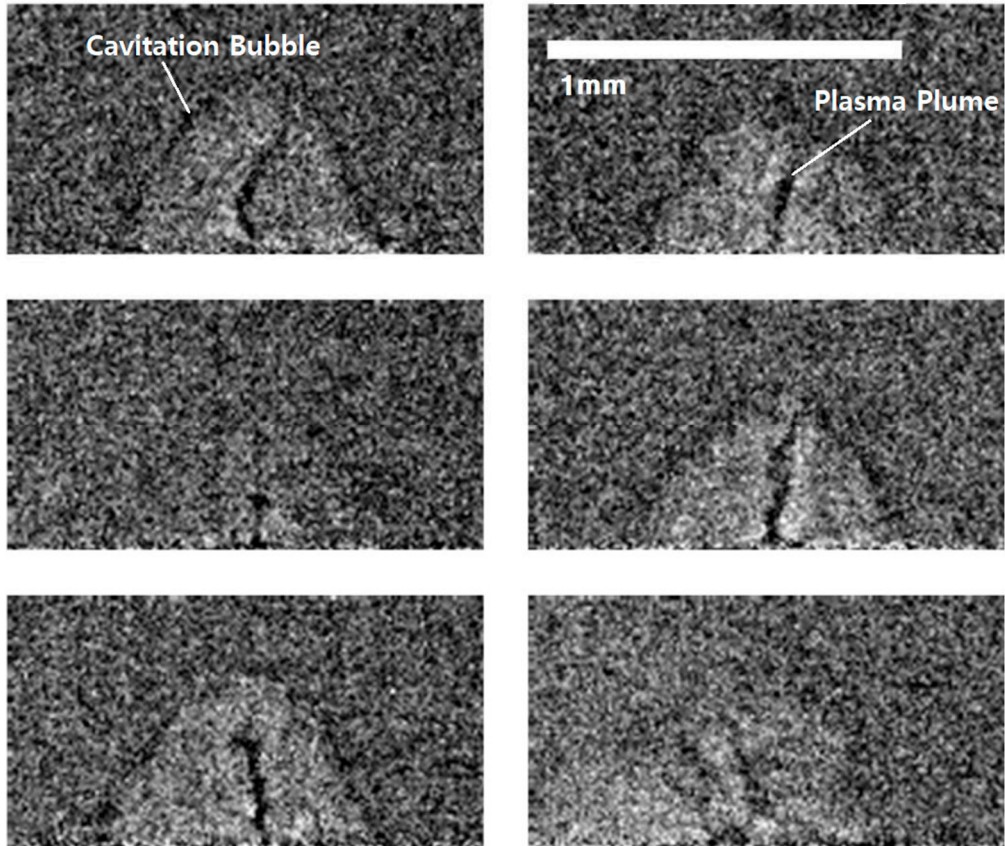

**Figure 4.** Representative individual radiographs from single laser shots at a fixed delay of ca. 320 microseconds, shows the variability of the plasma plume of phase VI (cavitation bubble collapse) (Reused with permission from ref. [50], copyright (2015), Nature).

### 3.2. Prevention of Cavitation Bubble

The temperature at the center of the high-power laser beam is 2200–3000 K [57,58], sometimes above 4000 K [45,55,59,60], and plasma is formed at the beam focus [50]. Occasionally, a low absorptivity material melts slightly even in an intense laser beam at first. Still, the melt is suddenly superheated as the absorptivity rises rapidly in a locally melted area. For example, the absorptivity of an aluminum alloy is only 5–15%, but that of steel exceeds 40% in an Nd:YAG laser with a one-micrometer wavelength [61]. Therefore, it is important to work rapidly with a high energy density for aluminum alloys [62].

The absorptivity of laser energy is related to the material composition, the incident laser angle, and the laser wavelength [61]. The angle is the relative range between the laser nozzle and the base metal. The energy wave (P wave) of a laser is absorbed only 25% at an incident angle of 70° to a ferrous material between the laser nozzle and the base metal. On the other hand, the absorptivity increases significantly to 70–82% at 87° or higher, and a one-micrometer wavelength Yb-doped fiber laser has high absorptivity in the 70–85° range. The ten-micrometer wavelength $CO_2$ laser has the highest absorptivity at 85–88.5° [61].

Therefore, the energy density can be adjusted by changing the incident laser angle [63]. In this adjusted area, the laser absorptivity changes due to the anisotropic absorptivity of the base metal surface [64], and cavitation is suppressed. Another way is to adjust the laser beam scanning velocity [65] or focus position [66–68], and a pulsed laser is desirable to change as a top-hat mode or a continuous wave laser (CW laser).

### 3.3. Prevention of Keyhole

A keyhole often forms at high scanning velocities and high ambient pressures [69] when the focus is deep with high power or a Gaussian beam. When the laser energy density

is higher than the thermal conductivity, keyholes are formed by an interaction between the absorbed laser energy and the free electrons of the metal. At this time, the electrons of the valence band transfer to the conduction band, and the migrating electrons interact with the metallic bonds, lattice strain, defect, imperfection, and potential perturbation [70]. A CW laser with a top-hat profile [71] or rectangular beam [72] laser is better for preventing keyhole formation.

Katayama's illustration and Kaplan's diagram are useful for understanding keyhole formation. Katayama reported that a keyhole is formed by recoil pressure due to evaporation [70], and Kaplan described in detail the keyhole in seven phases: (a) Melt flow redirected to pass around the keyhole, (b) Marangoni flow driven by surface tension gradients, (c) redirected flow that can cause spatter, (d) humping caused by accumulating downstream flow, (e) stagnation point for accelerated flow, (f) inner eddy, and (g) keyhole front melt film flow downwards by boiling recoil pressure, which is illustrated in Figure 5 [73].

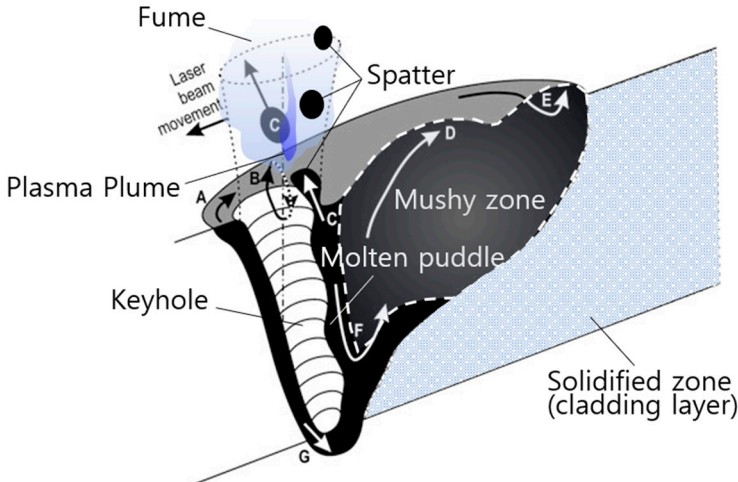

**Figure 5.** Modified schematic diagram of Kaplan's and Katayama's illustration, depicting the evaporation products formation by laser [70,73] ((Reproduced with permission from ref. [73], copyright (2017), Springer Nature)).

According to Khairallah et al., even in the shallow deposition of a PBF simulation, recoil pressure occurs similar to a keyhole because of the high temperature, and strong Marangoni convection occurs due to the surface tension during cooling [59]. Gunenthiram et al. examined spatter generation with a high-speed camera to understand better the spatter generation process according to the power change [62]. Unlike the laser welding process, the thin-DED process needs to reduce keyhole and spatter formation by avoiding a pulsed wave or Gaussian beam that makes deep and narrow melt pools.

### 3.4. Position of the Laser Beam Focus

The focus of the laser beam is influential because it is the highest energy area. When the defocus distance increases by moving the beam's focus slightly up or down on the base metal surface, the energy density of the melt pool decreases because the laser irradiation area widens [67]. Moreover, powder melt temperature decreases, which suppresses the generation of evaporation products.

Figure 6 shows the defocus in laser drilling, which explains the effect of the beam focus due to a focus shift and angle in Figure 6a–i [53,74]. For examples of Figure 6h,i, the influence appears similar to the defocus when the laser irradiation angle is changed. Li et al. reported that spatter was suppressed in the keyhole of laser welding when the focus position moved downward from the base metal surface [69].

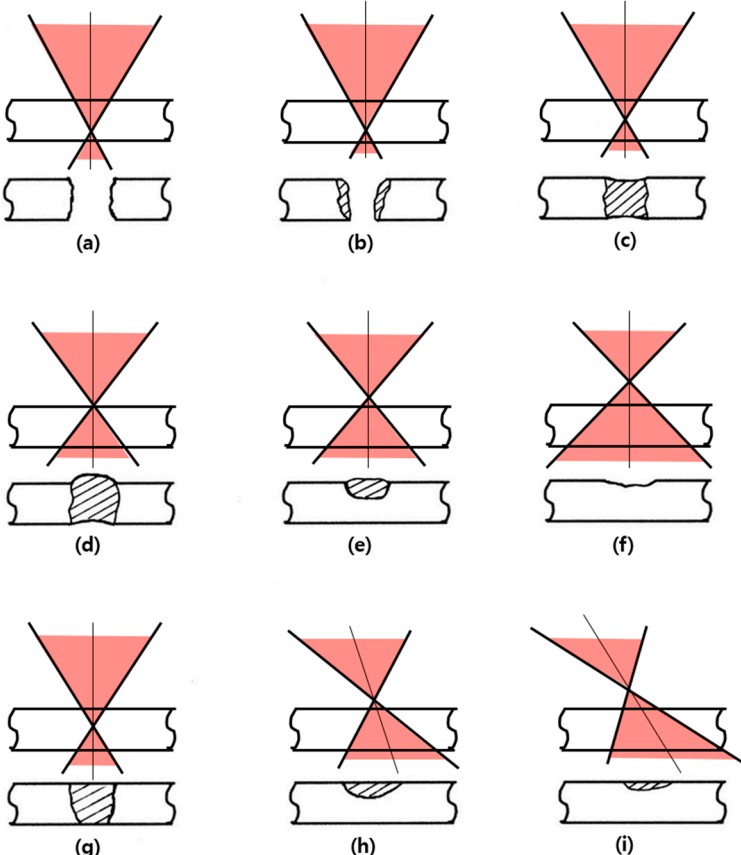

**Figure 6.** Effects of the beam focus position and beam angle on the focus depth and shape of laser drilling (**a–i**) (modified from Yeo's illustrations) [53,74].

In thin-DED, if the focus is on the surface of the base metal, the depth of the melt pool may become too deep due to the high laser power density, which may damage the original shape. On the other hand, it may leave an incomplete deposition layer and pores when the laser power is decreased to avoid overheat. The deep melt pool problem can be solved in two ways. First, it decreases the depth of the melt pool shallower by applying a defocus while maintaining power. Second, when the powder is supplied to the upper point of focus, the temperature of the laser beam focus decreases, which lowers the melt pool temperature of the base metal.

## 4. Influence of Powder Characteristics

Powder characteristics such as the size, chemical composition, and surface condition of the powder particles have a great influence on moisture absorption from the atmosphere and reaction with the laser beam, so it is important to select a suitable particle for the DED process to prevent evaporation products.

### 4.1. Powder Size and Chemical Composition

The powder size and chemical composition are important factors. The small powder has high surface energy and a short time for melting, which can be superheated easily. Porous sintered powder or fine powder has a large specific surface area to volume ratio. Thus, the amount of atmospheric moisture adsorbed and the laser absorptivity is relatively high. Fine particles under five micrometers in the powder cannot settle on the melt pool and are blown away by the gas flow or often meet the laser beam to become a fume. Therefore, a particle size of 30–45 μm is suitable for the high-power thin-DED process. The powder should have a similar thermal expansion coefficient to the mixed phases and a small difference in particle size distribution.

Because a laser is an electromagnetic wave, the lattice [70,75] condition and electron orbitals [76,77] vary according to the composition and absorptivity of the powder. Powders containing metal elements with a low evaporation point below 2000 °C or containing carbon or boron generate evaporation products easily and hinder the absorption of a laser on the base metal. When the chemical composition changes due to fume generation, it alters the melt fluidity, and thermal conductivity may cause the formation of pores or cracks in the coating layer. The compound-forming refractory components, niobium, tantalum, tungsten, and zirconium, are useful for improvement because they have vaporization points of 3000 °C or higher.

Evaporation is not determined merely by the melting point and vaporization point of the component. It depends on the eutectic reaction among atoms in the alloy or the formation of an intermetallic compound and the vapor pressure of the evaporated component. As an example, Figure 7 presents a comparison graph for evaporation of 304 stainless steel [78], where Figure 7a is the respective evaporation curves of iron, manganese, chromium, nickel, and Figure 7b is the evaporation curves of the four elements in the alloy. Among the four elements, the vapor pressure of manganese is the highest, but the vapor pressure of manganese in the alloy is the lowest. According to Henry's law, the partial pressures determine the vaporization rate because the manganese content is less than 2% in 304 stainless steel. Iron and chromium are present in 70% and 18%, respectively. On the other hand, neither carbon nor boron is presumed to follow Henry's law.

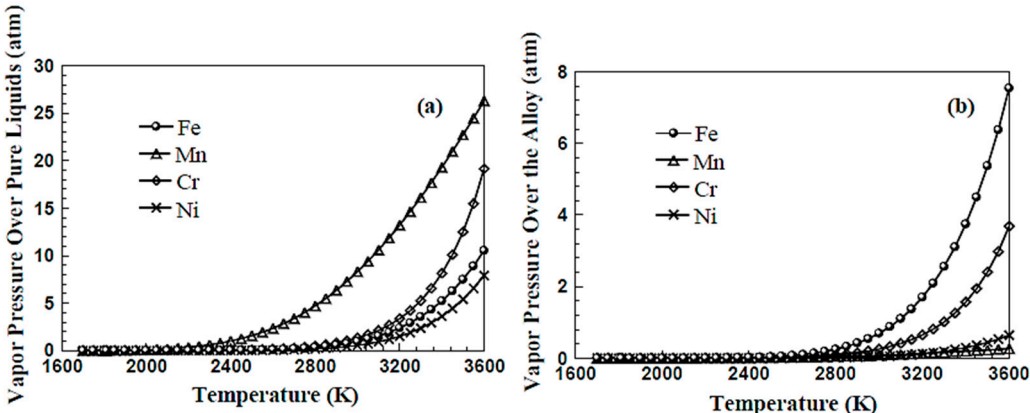

**Figure 7.** Equilibrium vapor pressures of the four alloying elements (**a**) over the respective pure liquids, and (**b**) over the alloy at different temperatures [78]. (Reused from ref. [78], copyright (2003), Sandia National Laboratories).

It is better to refer to the phase diagram rather than the Ellingham diagram to predict the presence of evaporation products. For example, regarding the Gibbs Energy of chromium carbides, many studies have reported that $Cr_{23}C_6$ is more stable than $Cr_7C_3$ or $Cr_3C_2$ in the Ellingham diagrams. On the other hand, these are different from the molten state because they present a solid-state range of 700–1300 °C. In the Cr-C phase diagram, however, the stability is $Cr_{23}C_6 < Cr_3C_2 \leq Cr_7C_3$ [79,80]. Berdnikov and Gudim and Vlasova et al. reported the evaporation point of chromium carbides in the order of $Cr_{23}C_6 < Cr_7C_3 < Cr_3C_2$ [81,82]. The powder containing chromium carbide has increased the vapor pressure of carbide when dissolved in 1527–1811 °C according to the C-Cr binary eutectic reaction [83] and Fe-Cr-C ternary eutectic reaction [84]. This presents the fume as oxidation [82] in the atmosphere. Despite this, there are few cases where refractory carbides, NbC, TaC, and ZrC, evaporate below 3000 °C [85].

*4.2. Surface Condition of Powder*

The composition and surface condition of powder is closely related to the laser absorption rate. The powder is produced mainly by four processes, crushing, atomizing, agglomeration, and sintering after agglomeration; the surface condition of the powder is

very influential in the thin-DED process. A hard and brittle powder could be obtained by crushing a bulk material. That is not used in the additive manufacturing process because of its polygonal and sharp edges. The alloy powder is manufactured mainly by atomizing, involving a molten metal spray with 10–50 MPa high-pressure gas or water. High melting point refractory carbide powders are usually produced by agglomeration with fine metal powders as binders and sintered to increase the density. Figure 8 shows the difference in surface morphology between the atomizing powder and agglomerated-sintering powder. The point to consider in powder selection is porosity in agglomerated powders. Another is a brittle phase by carbide dissolution in the binder metal [86]. Recently, plasma-atomized tungsten carbide powder [87–90] was used to prevent embrittlement of the cladding layer.

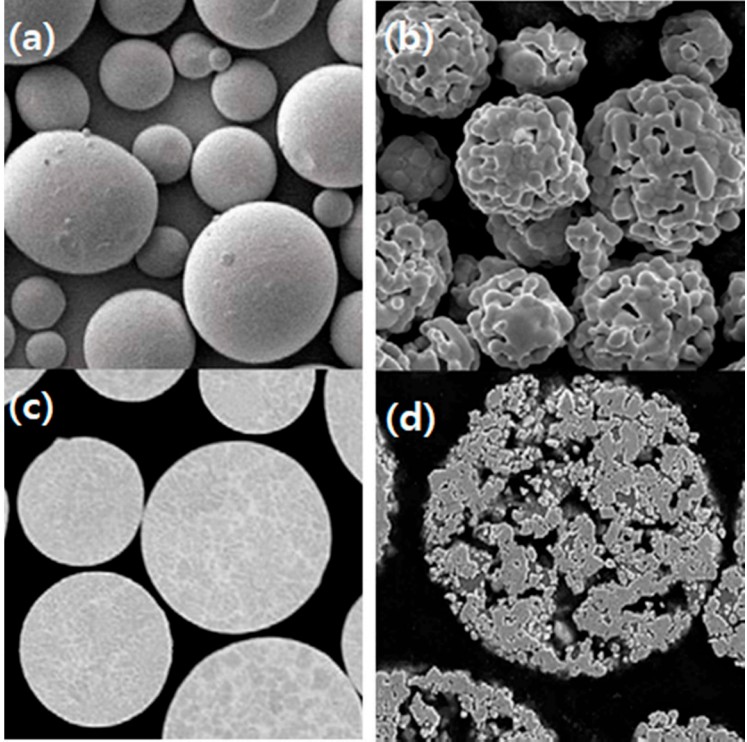

**Figure 8.** Powder morphology depicted in different manufacturing processes: (**a**,**b**) External surface; (**c**,**d**) Cross-section; (**a**,**c**) Metal alloy powder as gas atomized; (**b**,**d**) Regular carbide powder as agglomerated and sintered) [91]. (Reused from ref. [91], copyright (2018), Oerlikon Metco).

The wettability between the pair of base metal and deposition powder also has an influence. Insufficient wettability causes non-uniform deposition because of the different absorptivity between the deposited melt and the uncoated area. If the main compositions between the base metal and the deposition material are similar, the wettability by the similarity in physical properties is sufficient. When selecting a deposition material with a different main component from the base metal, a material with a similar thermal expansion coefficient is chosen.

The last factor is moisture adsorbed on the surface. Agglomerated and sintered powders or fine powders have high surface energy so that moisture can be adsorbed on the particle surface. Strongly adsorbed moisture is difficult to remove by heating or vacuum [92], and the surface generally tends to lower the surface energy by forming an oxide layer with moisture [93]. Most moisture adsorption is physical adsorption caused by weak van der Waals forces. Water molecules are difficult to remove by heating when chemical adsorption occurs by covalent bonding or ionic bonding with the reactive gases in porous powders or on the rough base metal surface [94,95].

As shown in Figure 9, chemical adsorption occurs because of the "image force", in which electrons on the oxygen atom side of the water molecule repel some electrons on the

powder surface and attract positive charges [96,97]. The adsorbed moisture reacts rapidly during powder melting and produces a fume containing oxides [15,98] and nitrogen oxides [99–103] that are harmful to the human body. Before use, powders should be dried in a heated oven for eight hours above the boiling point of water (100 °C), cooled, and immediately charged into the hopper to avoid contact with the atmosphere.

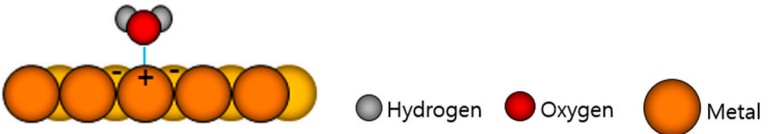

**Figure 9.** Example of a moisture particle adsorbed on metal by image force [97].

## 5. Process Parameters

### 5.1. Powder Feed Rate

Determining the suitable powder feed rate requires experiments. If the powder feed rate is too low, the melt pool is superheated and generates keyholes and cavitation bubbles. In contrast, if the powder feed rate is too high, the base metal is not heated uniformly, so the melt pool is not uniform and does not deposit appropriately.

For the experimental sequence, a specific range of the powder feed rates to the volume ratio was set, and the other parameters (power, scanning velocity, and hatch distance) were adjusted. Finally, the powder feed rate was fine-tuned to the desirable level.

### 5.2. Scanning Velocity

When increasing the scanning velocity on the base metal, the lack of powder should be considered. An increase in scanning velocity under the same powder feed rate means much more laser energy is supplied to the powders and base metal. Thus, base metal and powders can be superheated.

To maintain the powder feed rate while increasing the scanning velocity, the laser power must be reduced to achieve a constant energy density per powder unit weight. Indeed, when the scanning velocity increases, the deposited layer is better with an increase in the powder feed rate or a decrease in power density [104] because cavitation bubbles can form when the maximum temperature of melt increases. A uniform and thin deposited layer can be obtained under properly controlled process parameters, even high-speed thin-DED.

### 5.3. Powder Supplying Position

Unlike the usual DED of supplying powder to a focus, in thin-DED, the powder reacts with the laser beam first by supplying powder slightly above the focus or by a defocus. The remaining laser energy then reaches the base metal surface [105,106]. Thus, the laser energy to the melt pool is reduced, resulting in a thinner bonding zone, improved powder recovery, and faster deposition velocity.

### 5.4. Shape of the Laser Beam

The latest technology modulates a rectangular laser beam through a cylindrical lens array [107] or crossed Powell lenses [72]; combining it increases the productivity of the laser deposition process. A wide rectangular beam can allow uniform heating [108–111] compared to a Gaussian beam. Thus, the structure of the deposited layer has a uniform composition and dispersion strengthening by fine precipitates.

### 5.5. Environmental Parameters

Generally, the generation of evaporation products is accelerated if the thin-DED process is performed in an air environment. If the process atmosphere contains reactive gases, such as oxygen [112], nitrogen [99], sulfur [98,100], halogen group elements [113,114], or moisture [115–117], it accelerates the generation of evaporation products. In particular,

nitrogen gas sometimes promotes the oxidation of carbide [116,118] and the formation of nitrogen oxides [101] and nitrides [102]. The nitrogen radicals decomposed by the high energy of the laser beam become nitrogen oxides (NO$_x$) [99,100], and the nitrogen oxides in contact with the laser beam again decompose to become reactive oxygen species and nitrogen oxide ions [103]. These reactive oxygen species oxidize the carbide of the powder. Therefore, when fumes are generated continuously, a shielding gas should be changed to an inert gas, argon, or helium without nitrogen. Due to an excessive shielding gas flow forming turbulence, the amount of laminar flow gas is appropriate for covering the laser and melt pool [119].

### 6. Experimental Sequence of Thin-DED

Thin-DED experiments can be performed in the sequence of a single-track and a multi-track. In the single-track experiment [120], the possibility of bonding between the base metal and the deposited powder could be investigated first, except for the effects of overlap, preheating, and post-annealing. The suitability of powder selection or process conditions can be determined by examining whether defects are generated according to the reaction between the powder and the laser beam.

When evaporation products form, the cause of the defect can be analyzed using the Ishikawa diagram. Figure 10 is a remake to prevent evaporation products in the thin-DED process by modifying the diagram reported by Yeo et al. [74]. Evaporation products can be avoided by choosing suitable material, further finely optimizing the parameters working condition, source laser, assist gas, and laser focusing.

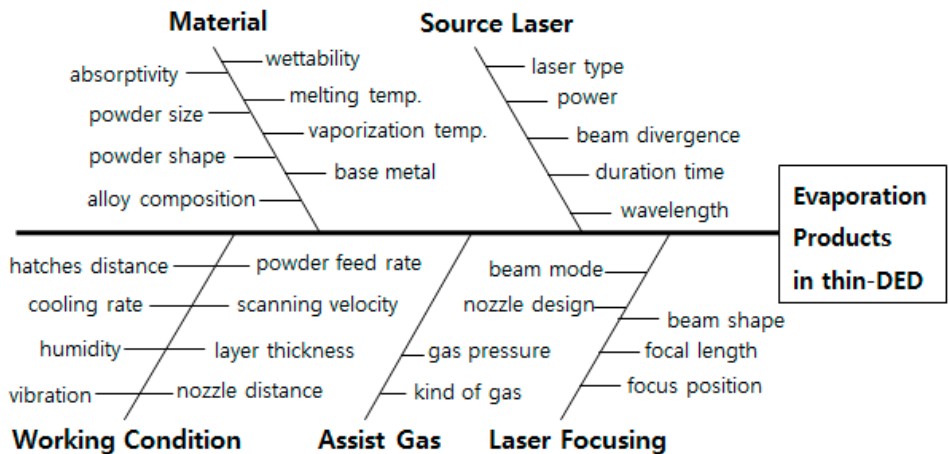

**Figure 10.** Ishikawa diagram showing the various parameters affecting the evaporation products in thin-DED (modified from Yeo's Ishikawa diagram) [74].

Figure 11 shows two experimental examples of single-track thin-DED. In Figure 11a, the laser power was low, and the base metal surface was not heated sufficiently. The wettability of the powder was also insufficient. Hence, the material did not deposit completely, and spatter was present at the beam boundary. Figure 11b gives an example of good results on a single-track. Therefore, an experiment to control the hatch distance in a multi-track experiment can produce a better-deposited layer than a single-track [97]. Figure 12 shows a thin-DED multi-track experiment on rolls.

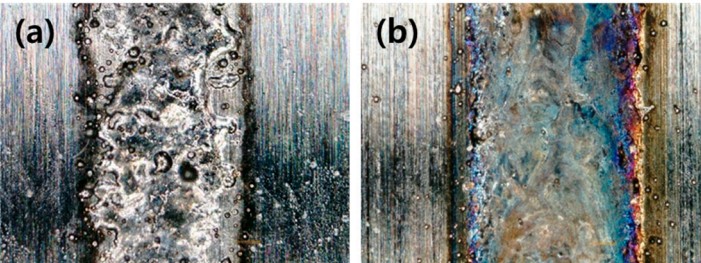

**Figure 11.** Single-tracks of different conditions between (**a**) incomplete single-track and (**b**) good single-track [97].

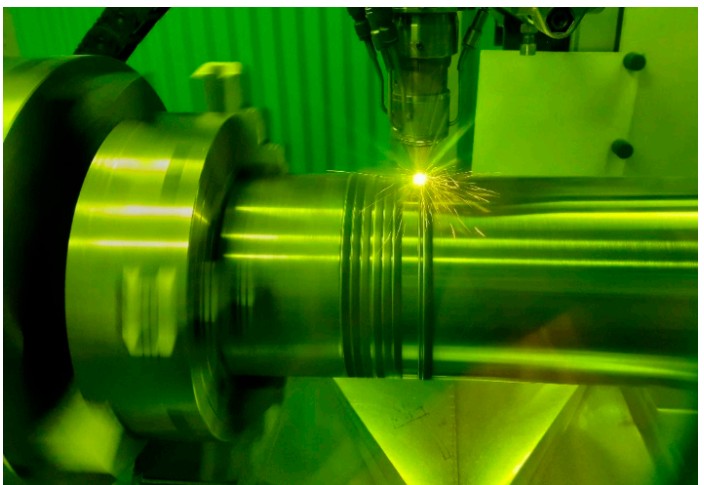

**Figure 12.** Photograph of a thin-DED test on a roller.

## 7. Summary

In thin-DED processes, nanoparticle fumes and massive agglomerated-spatter can be released when the base metal is heated at the focus to form a melt pool, which is collectively called evaporation products. Evaporation products can be prevented by selecting suitable powder and source laser, further finely optimizing process parameters. To prevent evaporation products in thin-DED, especially the following factors were investigated:

(1) To reduce the evaporation products, it is first necessary to suppress the concentration of the needle-like laser energy. Particularly a continuous wave laser, top-hat mode, and rectangular beam are more desirable than a pulsed laser or Gaussian beam.

(2) The defocus shifts the laser beam focus, widening the laser irradiation area and lowering the energy density.

(3) Fine particles with a size of less than five micrometers can form fumes easily. Powders with an average size of 30–45 micrometers are suitable in thin-DED.

(4) It is necessary to minimize carbon or boron and low evaporation point components. A sintered powder with a porous surface generates evaporation products because of high moisture adsorption and laser absorption in the air. Therefore, atomized powders and plasma atomized carbide powders with smooth surfaces are preferred for thin-DED.

(5) The bonding zone can be thinner by feeding the powder slightly above the focus to lower the laser energy density delivered to the melt pool. Powder recovery and nozzle scanning velocity can also be increased.

**Author Contributions:** K.-H.K. contributed primarily to writing the original draft and compiled the manuscript and the data collection; C.-H.J. contributed to writing, drawing, and data curation; D.-Y.J. contributed to writing-outline of the paper, reviewing and editing, supervision; S.-K.H. administrated a work, conceptualization, reviewing and editing the manuscript. All authors have read and agreed to the published version of the manuscript.

**Funding:** This research received no external funding.

**Data Availability Statement:** Not applicable.

**Acknowledgments:** The authors appreciate the financial support for this work from Inha University.

**Conflicts of Interest:** The authors declare no conflict of interest.

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
