# Peer review of "Preventing Evaporation Products for High-Quality Metal Film in Directed Energy Deposition: A Review"

_metals, doi:10.3390/met11020353_

Round 1

Reviewer 1 Report

Dear Authors

The review paper presented by You is very interesting and well written. However some minor fault were found and before publication have to be corrected. Below are listed things to change:

1. Why the figure 2 is placed in paragraph 1 and its description in paragraph 3.1, this should be one after other?

2. Please explain why on figure 3 is t<0. How is possible that the time bellow 0?

3. Figure 4 are in low quality.

4. Between points 4 and 4.1 should be placed 2-3 sentences.

Author Response

The review paper presented by You is very interesting and well written. However some minor fault were found and before publication have to be corrected. Below are listed things to change:

We will explain each answer and corrections as follows.

  1. Why the figure 2 is placed in paragraph 1 and its description in paragraph 3.1, this should be one after other?

Answer: We corrected the errors and thus the Figure 2 moved to 3.1 according to your advice.

  1. Please explain why on figure 3 is t<0. How is possible that the time bellow 0?

Answer: We modified an explanation below ‘t < 0’ and added some black dots in the cavitation bubble according to your advice. In fact, the expression ‘t < 0’ means before laser penetration, and it follows the expression of original journal by Prof. Stauss with respect.

  1. Figure 4 are in low quality.

Answer: Unfortunately, the original photo is in low resolution as they are radiographic photos of small bubbles.

  1. Between points 4 and 4.1 should be placed 2-3 sentences.

Answer: Authors agree with your suggestion and the modification have been made into the manuscript. We added its description between points 4 and 4.1 according to your advice as follows:

Powder characteristics such as the size, chemical composition, and surface condition of the powder particles have a great influence on moisture absorption from the atmosphere and reaction with the laser beam, so it is important to select a suitable particle for the DED process to prevent evaporation products.

Thank you for your valuable comments. Your kind advice has made the quality of this manuscript improved. 

Reviewer 2 Report

Dear Editor: I would like to express my deep thanks for inviting me to review the manuscript ID: metals-1093030

Title:    Preventing Evaporation Products for High-quality Metal Film in Directed Energy Deposition: A Review

Authors: Kang-Hyung Kim, Chan-Hyun Jung, Dae-Yong Jeong and Soong-Keun Hyun

Comments:

Abstract:

Please rewrite the abstract according to present and future development.

Introduction part:

Please include conventional process and its drawback.

Results and discussion:

  1. Figure 4 is not clear please provide clear image.
  2. Please add few conventional technique results

Summary part:

Please rewrite the summary part including suggestions

RECOMMENDATION

After reviewing the enclosed manuscript for “Metals”, the present manuscript contains some kinds of scientific analysis but it is mandatory required to modify according to the preceding remarks. So, the manuscript can be accepted for publication after minor mandatory revisions have been made.

Author Response

Abstract:

Please rewrite the abstract according to present and future development.

Answer: Thank you for your valuable comments. We revised the abstract according to your advice.

Introduction part:

Please include conventional process and its drawback.

Answer: We added the intensive review on the conventional processes and their drawbacks in the paragraph and also modified two terms as more suitable according to your advice.

Figure 4 is not clear please provide clear image.

Answer: Unfortunately, the original photo is in low resolution as they are radiographic photos of small bubbles.

Please add few conventional technique results

Answer: According to your advice, we added few conventional techniques including plasma spray, HVOF spray, flame spray, and detonation-gun.

Summary part:

Please rewrite the summary part including suggestions

Answer: We added suggestions in the summary paragraph according to your advice.

Thank you for your valuable comments. We revised the abstract according to your advice.

Reviewer 3 Report

Dear authors, 

below please find the comments and suggestions.

The authors stated the following: "Thermal spraying techniques have been strong candidates to replace chromium plating for over 20 years, but they are still unsuitable because of their porosity and low bonding strength. DED can achieve high bonding strength similar welding to fix delamination failure, which is persistent in plated and thermal sprayed layers." I want to emphasize that, as far as thermal spraying is concerned, the above general statements of the authors should be modified, because there are many thermal spraying techniques. In some processes (for example, flame spraying with simultaneous or subsequent fusing), an excellent metallurgical bond is achieved, delamination does not occur, etc., i.e. coatings of high quality can be obtained. Therefore, I suggest reformulating above statements.

The abbreviations first mentioned in the text or in figures (AM, XCT, HAZ, NP-Figure 3, CW…) should be explained. Mostly these are common abbreviations, but it is still better to do so.

Figure 2 – Moten Metal to Molten Metal

The paper 48 (Ibrahimkutty et al., 2015) is cited in Figure 2, Line 65. Should it be numbered by 12, i.e. after the previously cited paper 11 in the text-Line 60?

The authors stated, "Precise control of the powder supply from the nozzle, laser power adjustment, and the location where the powder and laser meet are key parameters in the process." What about the scanning velocity?

Please use uniformly the following: 50 - 300 mm, 50 ~ 300 mm; thin-DED, Thin-DED

Figure 3 – Please be careful with times; there are some dots? instead of minuses?

Figure 2 is mentioned in two different paragraphs of the text. I suggest that the figure not be mentioned in line 58 (I don't think it is necessary), but that it be replaced from line 60 closer to line 145, where it is actually described in detail. In that way, the earlier problem with citing the literature 48 will be avoided. I have to mention that this is the situation in my downloaded pdf version from the submission system.

In the title of Figure 4, the six phases should be mentioned and marked in radiographs as well.

What is connection between the paragraph in Lines 211 to 221 and sentences in Lines 225 to 227?

Berdnikov and Gudim and Vlasova et al. - Line 272

Figure 8 – please use a, b, c and d; the lower, upper etc. is not understandable.

The reference 117 (Oerlikon Metco) is cited in Figure 8, Line 297. Should it be numbered by 89, i.e. after the previously cited paper 88 in the text-Line 292?

Is it possible to have reference 93 and then 118 cited? – Line 316. There are no references 94 to 117.

The same is valid for 13, 104 and 103, 105, 112 – 114. – Line 317

The reference 118 (Kim et al.) is cited in Figure 9, Line 322. Should it be numbered by ?, i.e. after the previously cited paper ? in the text-Line 317?

Please check the citing in the rest of the submission.

The following sentence, "If the process atmosphere contains reactive gases, such as oxygen [102], nitrogen [103], sulfur [104, 105], halogen group elements [106, 107], 359 or moisture [108-110]." seems to be incomplete.

I suggest that the Conclusion be extended so that all the influencing factors listed in Figure 10 are commented out. Optionally, the table in previous section could be added with comments on all influencing factors (e.g. what is better to prevent fume and spatter).

What is the relation of this submission with the paper: Kim, K-H.; Jung, C-H.; Jeong, D-Y.; Hyun, S-G. Causes and Measures of Fume in Directed Energy Deposition: A Review. Korean J. Met. Mater. 2020, 58, 383-396. [http://dx.doi.org/10.3365/KJMM.2020.58.6.383]

Can the authors prove that they are the experts in the topic reviewed (by the published papers in high-quality journal)? Only in this way, a critical review and analysis of previous research is possible.

I would like to ask the authors, if they get the opportunity from the editor, not to write only "done" or something similar in their responses. In addition, I would like to ask the authors to write in the response letter below each comment what the changed part of the paper looks like, and not to instruct me to look in the revised manuscript.

Author Response

  1. The authors stated the following: "Thermal spraying techniques have been strong candidates to replace chromium plating for over 20 years, but they are still unsuitable because of their porosity and low bonding strength. DED can achieve high bonding strength similar welding to fix delamination failure, which is persistent in plated and thermal sprayed layers."

I want to emphasize that, as far as thermal spraying is concerned, the above general statements of the authors should be modified, because there are many thermal spraying techniques. In some processes (for example, flame spraying with simultaneous or subsequent fusing), an excellent metallurgical bond is achieved, delamination does not occur, etc., i.e. coatings of high quality can be obtained. Therefore, I suggest reformulating above statements.

Answer: We modified the manuscript as follows. In the paragraph, we added description about the conventional processes and their drawbacks and modified terms as more suitable according to your advice.

  1. The abbreviations first mentioned in the text or in figures (AM, XCT, HAZ, NP-Figure 3, CW…) should be explained. Mostly these are common abbreviations, but it is still better to do so.

Answer: We added the full words of each abbreviations first mentioned in the text or in figures according to your advice.

  1. Figure 2 – ‘Moten Metal’ to ‘Molten Metal’

Answer: We corrected an explanation in Figure 2 ‘Moten Metal’ to ‘Molten Metal’.

  1. The paper 50 (old 48) (Ibrahimkutty et al., 2015) is cited in Figure 2, Line 65. Should it be numbered by 12, i.e. after the previously cited paper 11 in the text-Line 60?

Answer: We corrected the errors and the Figure 2 moved to 3.1 according to your advice.

  1. The authors stated, "Precise control of the powder supply from the nozzle, laser power adjustment, and the location where the powder and laser meet are key parameters in the process." What about the scanning velocity?

Answer: We added the ‘scanning velocity’ as one of the key parameters in the process, in the last sentence from the first paragraph in section 3.

  1. Please use uniformly the following: 50 - 300 mm, 50 ~ 300 mm; thin-DED, Thin-DED

Answer: Following your comment, we unify the terms used.

  1. Figure 3 – Please be careful with times; there are some dots? instead of minuses?

Answer: We modified an explanation below ‘t < 0’ and added some black dots in the cavitation bubble according to your advice. In fact, the expression ‘t < 0’ means before laser penetration, it follows the expression of original journal by Prof. Stauss with our respect.

  1. Figure 2 is mentioned in two different paragraphs of the text. I suggest that the figure not be mentioned in line 58 (I don't think it is necessary), but

Answer: Figure 2 moved to 3.1 according to your kind advice.

  1. In the title of Figure 4, the six phases should be mentioned and marked in radiographs as well

Answer: We added an explanation for phase VI in the title according to your advice, and comments marked in radiographs.

  1. What is connection between the paragraph in Lines 211 to 221 and sentences in Lines 225 to 227?

Answer: We modified the paragraph as connection between lines 224 to 237 (old lines 211 to 221) and lines 238 to 241 (old lines 225 to 227) according to your advice.

Previous sentences: For thin-DED, the original shape of the base metal can be damaged if the melt pool temperature is too high due to the high power and high velocity. On the other hand, a decrease in laser power may leave an incomplete deposition layer and pores.

Revised sentences: In thin-DED, if the focus is on the surface of the base metal, the depth of the melt pool may become too deep due to the high laser power density, which may damage the original shape. On the other hand, it may leave an incomplete deposition layer and pores when the laser power is decreased to avoid overheat.

  1. Berdnikov and Vlasova et al. - Line 272

Answer: We revised the manuscript line 290 according to your advice.

  1. Figure 8 – please use a, b, c and d; the lower, upper etc. is not understandable.

Answer: We modified caption and Figure 3 according to your advice.

  1. The reference 117 (Oerlikon Metco) is cited in Figure 8, Line 297. Should it be numbered by 89, i.e. after the previously cited paper 88 in the text-Line 292?

Answer: We rearranged all number of references in the correct order according to your advice.

  1. Is it possible to have reference 93 and then 118 cited? – Line 316. There are no references 94 to 117.

Answer: We rearranged these references in the correct order including line 333 according to your advice.

  1. The same is valid for 13, 104 and 103, 105, 112 – 114. – Line 317

Answer: We rearranged these references in the correct order including line 333 to 334 according to your advice.

  1. The reference 118 (Kim et al.) is cited in Figure 9, Line 322. Should it be numbered by?, i.e. after the previously cited paper ? in the text-Line 317?

Answer: We rearranged these references in the correct order including line 339 according to your advice.

  1. Please check the citing in the rest of the submission

Answer: Thanks for your kind advice, we rearranged all references.

  1. The following sentence, "If the process atmosphere contains reactive gases, such as oxygen [102], nitrogen [103], sulfur [104, 105], halogen group elements [106, 107], 359 or moisture [108-110]." seems to be incomplete.

Answer: We modified it to a complete sentence as follows:

If the process atmosphere contains reactive gases, such as oxygen [110], nitrogen [97], sulfur [96, 98], halogen group elements [111, 112], or moisture [113-115], it accelerates the generation of evaporation products. (line 376-378)

  1. I suggest that the Conclusion be extended so that all the influencing factors listed in Figure 10 are commented out. Optionally, the table in previous section could be added with comments on all influencing factors (e.g. what is better to prevent fume and spatter).

Answer: Your suggestion is crucial, such that we could improve our paper significantly. Reflecting on your comments, we added an explanation for Figure 10 and modified sentences to the summary 1) paragraph according to your advice as follows:

1) To reduce the evaporation products, it is first necessary to suppress the concentration of the needle-like laser energy among various methods. Particularly a continuous wave laser, top-hat mode, and rectangular beam are more desirable than a pulsed laser or Gaussian beam.

  1. What is the relation of this submission with the paper: Kim, K-H.; Jung, C-H.; Jeong, D-Y.; Hyun, S-G. Causes and Measures of Fume in Directed Energy Deposition: A Review. Korean J. Met. Mater. 2020, 58, 383-396. [http://dx.doi.org/10.3365/KJMM.2020.58.6.383]

Answer: It is our paper on the fume, a factor of DED defects in the authors’ previous investigation. Subsequent investigations have found that spatter also produced during evaporation is a major factor in pore generation.

  1. Can the authors prove that they are the experts in the topic reviewed (by the published papers in high-quality journal)? Only in this way, a critical review and analysis of previous research is possible.

Answer: The first author has worked for the long time and a lot of experience in HVOF thermal spray coating and plating in the application field. In addition, other authors including corresponding also have many years of experience in metal casting and surface modification and published lot of papers. We are now broadening the scope of our research field with additive manufacturing. The purpose of this manuscript is to provide the important knowledge on fume and spatter in AM field.

We are grateful for your valuable comments. Your kind advice has made the quality of this manuscript improved. 

Reviewer 4 Report

Paper: Preventing Evaporation Products for High-quality Metal Film in Directed Energy Deposition: A Review, present some results from literature with impact on additive manufacturing area and represent an important data base for further 3D printing considerations for new materials. 

By my point of view there are few remarks that can help the paper to become an article, many of them minor but few of them major: 

  • Paragraph from Line (L) 22 to 31 (especially L25) need one or more references 
  • L49: micrometers 
  • L36: 1,3 
  • L39: 6,7
  • L72: Fig. 1 need a reference 
  • L220: Fig6 Explanations for a)-i) 
  • L313: Fig.8 scales 
  • L528 and L588 ref 47 and 72 - eliminate underline 
  • L175: page 6 : Fig4 appear twice 
  • L596: arrange reference 
  • In text there are no 55,56,58 and 59 references - must appear in text also 
  • Ref. 57 is after 61
  • Ref 60 is after 69 ...... Please, as a major concern, verify and arrange in correct order the references in text according to their appearance, for a review article this aspect is quite important. 
  •  
  •  

Author Response

By my point of view there are few remarks that can help the paper to become an article, many of them minor but few of them major: 

  1. Paragraph from Line (L) 22 to 31 (especially L25) need one or more references ((L)

Answer: Thank you for your valuable comments. The description is written by authors based on our own experience in the industrial field. In addition, we cited more references related to harmfulness of the hexavalent chromium including your suggestion. The list of additionally cited references is as follows:

Baruthio, F. Toxic effects of chromium and its compounds, Biol. Trace Elem. Res. 1992, 32, 145-153. [https://doi.org/10.1007/BF02784599]

Sun, H.; Brocato, J.; Costa, M. Oral Chromium Exposure and Toxicity, Curr. Envir. Health Rpt. 2015, 2, 295–303. [https://doi.org/10.1007/s40572-015-0054-z]

  1. L49: micrometers-

Answer: We modified according to your advice.

  1. L36: 1,3- 

Answer: We modified according to your advice.

  1. L39: 6,7-

Answer: We modified according to your advice.

  1. L72: Fig. 1 need a reference- 

Answer: The figure 1 is an own drawing from authors with much detailed information.

  1. L220: Fig6 Explanations for a)-i)- 

Answer: We modified by inserting (a)-(i) in the text explanations as follows:

Figure 6 shows the defocus in laser drilling, which explains the effect of the beam focus due to a focus shift and angle in (a) – (i) [53, 74].

  1. L313: Fig.8 scales 

Answer: Unfortunately, also no scales in the original figures, it just a reference showing powder external surfaces and cross-sections.

  1. L528 and L588 ref 47 and 72 - eliminate underline –

Answer: We modified according to your advice.

  1. L175: page 6 : Fig4 appear twice –

Answer: We modified according to your advice.

  1. L596: arrange reference –

 Answer: We modified according to your advice.

  1. In text there are no 55,56,58 and 59 references - must appear in text also –

Answer: We modified according to your advice.

  1. Ref. 57 is after 61 –

Answer: We modified according to your advice.

  1. Ref 60 is after 69 ...... Please, as a major concern, verify and arrange in correct order the references in text according to their appearance, for a review article this aspect is quite important. 

Answer: We modified and rearranged all references in the correct order according to your kind advice.

Thank you for your valuable comments. Your kind advice has made the quality of this manuscript improved.

Round 2

Reviewer 3 Report

Dear authors, 

Thank you for your response.

Below are my comments from review 1, which you may not have understood and therefore did not respond to in full.

  1. Original comment from review 1: Figure 3 – Please be careful with times; there are some dots? instead of minuses?

I meant the following: what is, e.g. 10.4 (ten dot four)?

  1. Original comment from review 1: In the title of Figure 4, the six phases should be mentioned and marked in radiographs as well.

I meant the following: Figure 4. Representative individual radiographs from single laser shots at a fixed delay of ca. 320 μs, shows the variability of the material ejection; I – plasma formation, II – cavitation buble expansion, etc.

If you mean that it is not necessary to have explanations of phases in the title of Figure 4, at least the radiographs should be marked from I to VI (according to the text in the previous paragraph).

  1. Original comment from review 1: Berdnikov and Gudim and Vlasova et al. - Line 272

Unfortunately, in your response, you have modified my comment “Berdnikov and Gudim and Vlasova et al. - Line 272” to “Berdnikov and Vlasova et al. - Line 272”.

Let me explain – when there are two authors, it is common to write both of them; i.e. you will have Berdnikov and Gudim; please check other references citations in the text.

  1. Original comment from review 1: I suggest that the Conclusion be extended so that all the influencing factors listed in Figure 10 are commented out. Optionally, the table in previous section could be added with comments on all influencing factors (e.g. what is better to prevent fume and spatter).

It is nice of you to highlight five important factors, but it was assumed that you explain all the sub-factors in more detail in Conclusion. The second option was to add table with comments on all influencing sub-factors to previous section (e.g. what is better to prevent fume and spatter). Unfortunately, nothing was done.

  1. Moreover, for the version 2, the following suggestions are as follows:

5.1. It is uncommon to have a figure first, followed by the text in which the figure is explained. Figure 5 should be mentioned in the text before and not later. The same is valid for Figure 6 and Figure 7.

5.2. Figure 8 should be inserted after the first or second paragraph of the section 4.2., to have a continuation in citing references. Now, you have 90 and then 92.  

5.3. Dear authors, you have two very similar sentences: a newly added “Evaporation products can be avoided by choosing suitable material, further finely optimizing the parameters working condition, source laser, assist gas, and laser focusing.” (Lines 397 to lines 399) and a newly added „Evaporation products can be avoided by selecting suitable materials and source laser, and further by finely optimizing the parameters working condition, assist gas, and laser focusing. (Lines 418 to 420).

It is not common to have the same sentence in the Conclusion that has been used before.

5.4. In the first version, the source of Figure 11 was from the paper Kim, K-H.; Jung, C-H.; Jeong, D-Y.; Hyun, S-G. Causes and Measures of Fume in Directed Energy Deposition: A Review. Korean J. Met. Mater. 2020, 58, 383-396. [http://dx.doi.org/10.3365/KJMM.2020.58.6.383].

In the second version,  the source of the same Figure (Figure 11) is from another paper: Patalas-Maliszewska, J.; Feldshtein, E.; Devojno, O.; Sliwa, M.; Kardapolava, M.; Lutsko, N. Single-tracks as a Key Factor in Additive Manufacturing Technology-Analysis of Research Trends and Metal Deposition Behavior. Materials 2020, 13, 1115. [https://doi.org/10.3390/ma13051115].

Author Response

  1. Original comment from review 1: Figure 3 – Please be careful with times; there are some dots? instead of minuses? I meant the following: what is, e.g. 10.4(ten dot four)?

Answer: Thank you very much for your kind advice. We missed the minuses error in the explanation since the first author was only focusing on the drawing. We modified it as advised in our third version. The second revised parts are distinguished in blue letters.

  1. Original comment from review 1: In the title of Figure 4, the six phases should be mentioned and marked in radiographs as well.

I meant the following: Figure 4. Representative individual radiographs from single laser shots at a fixed delay of ca. 320 μs, shows the variability of the material ejection; I – plasma formation, II – cavitation bubble expansion, etc.

If you mean that it is not necessary to have explanations of phases in the title of Figure 4, at least the radiographs should be marked from I to VI (according to the text in the previous paragraph).

Answer: The explanation in our paper was insufficient. As shown in the following figure of cited paper [50], phase I through V corresponds to less than 300 microseconds. In the cited paper [50], Figure 4 is radiographs of phase VI at 320 microseconds, where the plasma is generated within the cavitation bubbles. Followed by the plasma plume rising and then bubble collapses. Thus, we have revised lines 158-160 to read the explanation in the paragraph on Figure 4 according to your advice.

Ibrahimkutty et al. observed the moments at 320 microseconds (phase VI) in which the bubble collapse after the rise of the plasma plume inside the cavitation bubble, as shown in Figure 4 by X-ray [50]. At that moment, the evaporation products are released into the atmosphere.

Figure 2. (a) Averaged X-ray radiographs of the bubble kinetics at selected delays after laser impact, some critical frames are highlighted: the largest extension of the first bubble at 110 μs; shrinking of the first bubble at 156 μs; second bubble (rebound) at 227 μs; jet formation after the second bubble has collapsed at 320 μs. Brighter pixels correspond to higher X-ray transmission. (b) plot of the lateral and vertical bubble radius as function of delay. Error bars are in both cases similar. The solid line is a simulation according to the Rayleigh-Plesset equation. (c) Mass of ablated particles at a height of 0.5 mm as function of delay. The line indicates the bubble size change according to (b). The arrows mark the rim. The contact angle is indicated by α.

Figure 3. Representative individual radiographs from single laser shots at a fixed delay of ca. 320 μs showing the variability of the material ejection. The scale bar is 1 mm.

[50. Ibrahimkutty, S.; Wagener, P.; Rolo, T.S.; Karpov, D.; Menzel, A.; Baumbach, T.; Barcikowski, S.; Plech, A. A hierarchical view on material formation during pulsed-laser synthesis of nanoparticles in liquid. Sci. Rep. 2015, 5, 5.]

  1. Original comment from review 1: Berdnikov and Gudim and Vlasova et al. - Line 272

Unfortunately, in your response, you have modified my comment “Berdnikov and Gudim and Vlasova et al. - Line 272” to “Berdnikov and Vlasova et al. - Line 272”.

Let me explain – when there are two authors, it is common to write both of them; i.e. you will have Berdnikov and Gudim; please check other references citations in the text.

Answer: Thank you for your kind advice, and we have corrected authors as advised.

Berdnikov and Gudim and Vlasova et al. reported the evaporation point of chromium carbides in the order of Cr23C6<Cr7C3<Cr3C2 [81, 82].

  1. Original comment from review 1: I suggest that the Conclusion be extended so that all the influencing factors listed in Figure 10 are commented out. Optionally, the table in previous section could be added with comments on all influencing factors (e.g. what is better to prevent fume and spatter).

It is nice of you to highlight five important factors, but it was assumed that you explain all the sub-factors in more detail in Conclusion. The second option was to add table with comments on all influencing sub-factors to previous section (e.g. what is better to prevent fume and spatter). Unfortunately, nothing was done.

Answer: Following your heartfelt advice, we have completed and added the following table:

Table 1. Preventing of DED defect due to fume and spatter

Related DED defect

Major factor

Sub-factor

Preventive measure

lower corrosion resistance, decreased fatigue strength, inner crack, surface crack, surface dimple, porosity, high roughness, uneven hardness, decreased laser energy efficiency, lower fluidity of melt.

material

adsorptivity, small powder size (<15㎛), low melting point or low vaporization temperature (<2000℃) component, contained carbon or boron, eutectic reaction, low wettability, characteristics of base metal.

adjustment of laser power, irradiation angle, surface condition of the powder and base material.

selection of suitable powder size, change material to higher vaporization temperature element, very low carbon and boron content.

source laser

laser type, laser power density, beam divergence, duration time, wavelength of laser.

adjustment of laser power, powder feed rate.

selection of CW laser, top-hat mode beam, bigger diameter beam, rectangular beam, or defocusing.

working condition

hatches distance, powder feed rate, scanning velocity, nozzle distance, cooling rate, layer thickness, humidity, vibration.

establishment again from single track experiments.

adjustment of hatches distance, powder feed rate, scanning velocity, nozzle distance, cooling rate, and layer thickness.

use air conditioner, and vibration absorber.

assist gas

gas pressure, kind of gas.

adjustment to the suitable pressure for complete air shielding. 

change to pure inert gas (argon or helium mixed argon).

laser focusing

beam mode, beam shape, nozzle design, focal length, focused on the base material surface.

selection of top-hat mode beam, bigger diameter beam, rectangular beam, defocusing, inclined laser beam.

  1. Moreover, for the version 2, the following suggestions are as follows:

5.1. It is uncommon to have a figure first, followed by the text in which the figure is explained. Figure 5 should be mentioned in the text before and not later. The same is valid for Figure 6 and Figure 7.

5.2. Figure 8 should be inserted after the first or second paragraph of the section 4.2., to have a continuation in citing references. Now, you have 90 and then 92.

Answer: We have moved the figures according to your kind advice.

5.3. Dear authors, you have two very similar sentences: a newly added “Evaporation products can be avoided by choosing suitable material, further finely optimizing the parameters working condition, source laser, assist gas, and laser focusing.” (Lines 397 to lines 399) and a newly added „Evaporation products can be avoided by selecting suitable materials and source laser, and further by finely optimizing the parameters working condition, assist gas, and laser focusing. (Lines 418 to 420).

It is not common to have the same sentence in the Conclusion that has been used before.

Answer: Following your kind advice, we have modified the sentence at the summary as follows:

Evaporation products can be prevented by selecting suitable powder and source laser, further finely optimizing process parameters. To prevent evaporation products in thin-DED, especially the following factors were investigated:

5.4. In the first version, the source of Figure 11 was from the paper Kim, K-H.; Jung, C-H.; Jeong, D-Y.; Hyun, S-G. Causes and Measures of Fume in Directed Energy Deposition: A Review. Korean J. Met. Mater. 202058, 383-396.

[http://dx.doi.org/10.3365/KJMM.2020.58.6.383].

In the second version,  the source of the same Figure (Figure 11) is from another paper:

Patalas-Maliszewska, J.; Feldshtein, E.; Devojno, O.; Sliwa, M.; Kardapolava, M.; Lutsko, N. Single-tracks as a Key Factor in Additive Manufacturing Technology-Analysis of Research Trends and Metal Deposition Behavior. Materials 2020, 13, 1115. [https://doi.org/10.3390/ma13051115].

Answer: Thank you for your kind advice, and we have corrected the cited reference numbers as advised. It was typos while revising the bibliography.

Reviewer 4 Report

I agree with the publication in this form

Author Response

Thank you very much for your kind advice.

We appreciate it gratefully.

Round 3

Reviewer 3 Report

Dear authors, 

Thank you for your response.

Some efforts are done, but the crucial newly added Table 1 could be of higher quality. In fact, mostly you have rewritten the sub-factors from Ishikawa presentation. Even opposite claims can be found (low vaporization temperature; change material to higher vaporization temperature element).  Furthermore, English language and style in this Table is very poor, i.e. completely different from the rest of the text. Obviously, you did not have enough time to be more precise.

In my opinion, authors should, in order to save their time and the time of reviewers and editors, respond carefully to all comments and suggestions for the first time.

Author Response

Answer: We are deeply sorry for disappointing your expectations due to insufficient English language skills. While the first author was revising the manuscript, we suddenly received a withdrawn email from the editor and sent the revised manuscript in a hurry, so we did not have plenty of time to edit the second version manuscript together. The withdrawn email was found to be an incident that occurred due to duplicate submission as the revised manuscript was uploaded. Now the incident has been handled.

We followed your last advice that “the second option was to add table with comments on all influencing sub-factors to previous section” and accordingly we inserted this Table 1 at the end of Section 1. Hopefully, the sections after this Table 1 will be in a sequence that explains the major factors and sub-factors in detail, making it easier for the reader to understand.

The awkward English expressions in the table have also been corrected according to your comments. Also, if the sub-factor contains either low melting temperature component, low vaporization temperature component, carbon, or boron, then fume or spatter is likely to be generated. This is described in lines 265-267 of the third-version manuscript (in lines 260-263 of the second-version manuscript).

Lastly, a phrase in the table is revised: the “change material to higher vaporization temperature element” of the preventive measure was changed to “addition of compound-forming refractory components (Nb, Ta, W, Zr)” according to your advice. Please open the attached file.
